# Overweight Prevalence among Rural Adolescents by Household Head Obesity and Socio-Economic Status in Limpopo, South Africa

**DOI:** 10.3390/children9111728

**Published:** 2022-11-10

**Authors:** Peter M. Mphekgwana, Masenyani O. Mbombi, Livhuwani Muthelo, Joseph Tlouyamma, Rathani Nemuramba, Cairo Ntimana, Katlego Mothapo, Inos Dhau, Eric Maimela

**Affiliations:** 1Research Administration and Development, University of Limpopo, Sovenga St, Polokwane 0727, South Africa; 2Department of Nursing Science, University of Limpopo, Sovenga St, Polokwane 0727, South Africa; 3DIMAMO Population Health Research Centre, University of Limpopo, Sovenga St, Polokwane 0727, South Africa; 4Department of Geography and Environmental Studies, University of Limpopo, Sovenga St, Polokwane 0727, South Africa; 5Department of Public Health, University of Limpopo, Sovenga St, Polokwane 0727, South Africa

**Keywords:** overweight, household head, blood pressure, adolescents, rural

## Abstract

Background: Childhood obesity has become the most important and growing public health problem in the world. They add to public health challenges by increasing the burden of chronic non-communicable diseases. However, in spite of its importance, there is limited literature that evaluates the prevalence of obesity among rural adolescents in sub-Saharan Africa. We report the first study to present an insight into rural black overweight South African children and the physical characteristics and socio-economic status of the household head. A quantitative cross-sectional population study was conducted involving 51 selected primary villages within the DIMAMO surveillance area in the Capricorn District of Limpopo Province, South Africa. The study involved 294 adolescents, 154 girls and 140 boys, who were under the age of 18. Of these participants, 127 (43%) were within the normal weight range, and 167 (57%) were overweight. Gender made a significant difference, with more girls being overweight than boys. Adolescents who did not receive child grants and whose heads of household were 45–54 years old, poor, and overweight had a higher prevalence of obesity (*p*-value < 0.05). This study suggests that public health interventionists need to target both the heads of household and their children in hopes of reducing the prevalence of overweight and obese South African children. We further propose a better understanding of the causes of childhood overweight and obesity to guide policy development and implementation in rural settings.

## 1. Introduction

Being overweight and obese has become the most important and growing public health concern for children around the world. They add to the public health challenge by increasing the burden of chronic non-communicable diseases (NCDs), which affect both developing and developed countries, including South Africa [1,2]. The prevalence amongst children aged 5 and 19 years has increased from 30.3 million in 2000 to 38.3 million in 2019 [3]. It is a multifactorial condition marked by excess adiposity caused by a complex interaction of biological, developmental, genetic, and environmental factors [4,5,6]. Obesity and overweight in children are on the rise in low- and middle-income countries, particularly in urban areas, with about a quarter of African children under the age of five being overweight. The primary cause of this in children is diet-related factors, including unhealthy food choices, such as high-energy foods, and also a lack of physical activity and/or a sedentary lifestyle [4,7,8,9].

The significant effects of obesity on children are grouped as physical health, social, psychological health, educational, and emotional well-being, such as developing early puberty in children, menstrual irregularities in adolescent girls, sleep disorders such as poor self-esteem, body image, and peer relationships, eating disorders, poor academic performance, and a lower quality of life [10]. Furthermore, Nicolucci and Maffeis (2022) noted that obesity in children represents a strong predictor of the condition and higher mortality in adulthood [11]. Moreover, children who grow up in poverty are 1.5 times more likely to be overweight and 1.6 times more likely to be obese [12], and the adverse effects of poverty and being overweight or obese have a far-reaching impact on educational outcomes.

Given the above-mentioned significant effect of obesity on children, there is a need for implementing effective treatments. So far, the effectiveness of obesity interventions has been based on health screening, individual behavioral changes, increasing daily physical exercise, or improving the quality of diet by restricting excess calorie intake [4,11,13]. For instance, health screening, a medical procedure performed to assess the likelihood of having a particular problem such as obesity in children [14], is used to determine if a child is obese or not by studying their BMI. On the other hand, prevalence studies could be used to inform researchers, guideline developers, and policymakers about the burden of obesity and the effectiveness of treatment for a targeted population [15].

However, there is limited literature in sub-Saharan Africa (SSA) to assess the prevalence of obesity among rural children by education level, employment, and the obesity status of the head of household in the rural setting. This burden of obesity in children is not foreign to South Africa, where the prevalence of overweight in children (aged 14–17) of African ancestry has been reported to be between 30% and 40% [16]. In the study reported here, we surveyed 294 adolescents with their heads of households to discover characteristics of Limpopo household heads related to obesity in rural-dwelling black South African children. We sought to deepen our understanding of the causes of being overweight and obese in children in order to guide policy development and the implementation of interventions to reduce the prevalence of overweight in rural children.

## 2. Methodology

### 2.1. Study Design

A quantitative cross-sectional population study was conducted within a bigger ongoing project of the DIMAMO Health and Demographic Surveillance Site (HDSS) involving 51 selected primary villages within the Dikgale, Mamabolo, and Mothiba (DIMAMO) surveillance area in the rural Capricorn District of Limpopo Province, South Africa. There were about 100,000 people under surveillance, mostly Northern Sotho speakers. However, there were few adolescents in these households. The study involved 294 adolescents, 154 girls and 140 boys, who were under the age of 18. Each household within the study site received a structured questionnaire that was developed, incorporated into the survey solutions, and distributed by trained field workers. All parents or guardians of the participants consented and the participants gave their informed consent in writing, while the Turfloop Research Ethics Committee (TREC) committees approved the study’s protocol. The tribal authority within the study setting authorized the study’s execution and provided written, informed consent.

### 2.2. Measurement

According to the International Society for the Advancement of Kinanthropometry protocol, each child underwent a series of anthropometric measurements [17]. With the subject in an anatomical position and the head in a Frankfort plane, height was measured with a Martin anthropometer to the nearest 0.1 cm. On electronic scales, bodyweight was calculated to the nearest 0.1 kg while wearing light clothing and no shoes. Body mass index (BMI) was determined by dividing weight in kilograms per square meter (kg/m^2^) by height. All anthropometric measurement training was carried out following the ISAK’s established protocols [17]. Demographic data for the participant, including age, gender, and educational attainment, were taken from the Dikgale HDSS database. The questionnaire was initially created in English, translated into the local language of Northern Sotho, and validated using construct, content, and face validity, as well as a pilot study.

### 2.3. Statistical Analyses

Statistical Package for Social Sciences (SPSS) version 26.0 software (I.B.M., Armonk, New York, NY, USA) was used to compute descriptive and inferential statistics. Data were reported as frequency and percentages. A comparison of categorical variables was performed using chi-square, and the statistical significance was set at *p* < 0.05 with a 95% confidence interval. Logistic regression analyses were used to determine the association between overweight adolescents with selected socioeconomic covariates and overweight status among the head of household.

## 3. Results

The characteristics of the participants are described in Table 1. Of the 294 participants, 127 (43.2%) were within the normal weight limit, and 167 (56.8%) were overweight. Most heads of household were under 30 years old (35%), followed by those aged ≥55 years. There were almost twice as many female (68%) household heads as males. More than half of household heads said they were just scarping by financially (65%) and more than 80% obtained less than a high school and were unemployed. Most adolescents were receiving child grants and living with their relatives. Around fifty-seven percent (57%) of household heads were overweight (Table 1).

There are no significant differences between the overweight status of the child and the prevalence of household head gender, education level, employment status, religious faith, and relationship with the child (*p*-value > 0.05). There were significant gender differences in terms of the prevalence of the child’s weight being within the normal range or overweight, with more girls being overweight than boys. Table 1 shows that adolescents who were not receiving a child grant, had an overweight head of household aged 45–54 years, and were poor had a higher prevalence of being overweight (*p*-value < 0.05).

The following factors were identified in the logistic regression model as being associated with overweight adolescents: (a) child’s sex, with boys having a lower risk than girls do (OR = 0.163; 95% CI: 0.067–0.396); (b) age of household head: adolescents with a household head aged between 45 and 54 years are at higher risk of being overweight than adolescents with a household head aged over 54 years (OR = 3.720; 95% CI: 1.203–11.501). In comparison to those who did not receive a child grant, those who received child grants had a lower risk of being overweight. The higher odds of adolescents who were overweight were found in households where the household head was overweight (Table 2).

The mean values of systolic and diastolic blood pressure with respect to age are shown in Figure 1 and Figure 2, respectively. As can be seen, mean systolic blood pressure was higher in adolescents who were overweight. Systolic blood pressure also decreased with age in both normal weight and overweight groups. However, there were no significant difference in the mean blood pressure (systolic and diastolic) between the normal weight and overweight groups.

## 4. Discussion

Sustainable development goal three in South Africa emphasizes establishing good health and well-being, including the health of children and adolescents. Hence, this study focused on the prevalence of obesity and its components among adolescents in the Dikgale Health and Demographic Surveillance System of Limpopo, South Africa (HDSS). The study further elucidated the odds ratio of overweight adolescents with selected variables of the household head. The prevalence of overweight children was 57%, with a high prevalence observed among those who are receiving child grants. Child obesity alone was reported to be less than 22.8% among South African children in 2010 [18]. The results of the current study show a rise in overweight rural children in the country. This is consistent with a recent study by Engwa et al. (2022) that revealed an increase in the prevalence of overweight in children [16]. However, the prevalence was high compared to studies in other sub-Saharan African countries such as Sierra Leone (16.9%), Comoros (15.9%), Malawi (14.5%), Ethiopia (3.0%), Togo (2.6), and Senegal (2.0%) [19]. The rise in prevalence is fuelled by etiological factors such as genetic, environmental, socio-economic risk, parental obesity, and parental income [20,21].

Child obesity is reported to differ between males and females [11,16,22]. The prevalence of being overweight was previously reported to be higher in girls than boys during adolescence, but there was no gender difference in early childhood [22]. This result is consistent with our findings, which showed that there were more girls than boys who were overweight and indicated that boys have a lower risk of being overweight than girls. This situation can be seen in the adult population as well, with women having the greatest likelihood of being overweight [23,24]. A study by Zong and his colleagues found that children raised in an extended family, mainly raised by their grandparents, or in high-income households may have an increased risk of becoming obese/overweight [25]. Contrary to other research [25], this study shows that adolescents staying with household heads between the ages of 45 and 54 are more likely to be overweight than adolescents living with household heads over the age of 54. The results’ discrepancy may be due to the socioeconomic disparities between the two nations.

A study by Shiba and Kondo looked into whether differences in the risk of becoming overweight between Japanese children from single-parent and two-parent families rose following the 2008 financial crisis [26]. In their study, they discovered that during the 2008 financial crisis, the risk of being overweight increased significantly for children from single-parent households. As a result, it was concluded that financial support and more childrearing support are needed to provide opportunities for single parents seeking jobs, social relationships, and support in raising their children. The current study showed a significant relationship between child grants and the weight status of children, with more adolescents who did not receive child grants being overweight. The Child Support Grant (CSG) is an important instrument of social protection in South Africa, reaching over 10 million South African children each month [27]. The South African CSG was first introduced in 1998. Early enrolment in the CSG reduced the likelihood of illness, with the effect being particularly strong for boys, particularly. It was further reported that receipt of the grant by adolescents generates a range of positive impacts, not the least of which is a reduction in risky behaviors [27].

Our results revealed that there was no significant difference in the systolic and diastolic blood pressure of normal weight and overweight children. However, systolic blood pressure was higher in adolescents who were overweight, and it decreased with age. Higher odds of overweight adolescents were found in households where the head of the household was overweight. A parent’s weight status has a significant influence on the risk of childhood obesity, and it was suggested that parents might play a crucial role in mitigating child obesity [2]. Underaged children are generally dependent on their parents’ choices of food. This is the case whether the food is healthy or not. Thus, the consumption of energy-dense food by parents may suggest the same for their children. Energy-dense food is one of the main contributors to obesity [20]. Parents lifestyle choices, such as a sedentary lifestyle and lack of physical activity, can influence those of their children. The type of food consumed by families is influenced by their lifestyle, affordability, and socio-economic status.

Childhood obesity associated with household socio-economic status such as low-income households was reported to be the leading factor of obesity among children [28,29,30,31,32]. The relationship between a parent’s or household’s socio-economic status and child obesity was stronger in Asia than in Europe and the Middle East, and it was stronger in high-income countries than in middle- or low-income countries [2]. However, our study, which was conducted in middle- or low-income countries, found no significant association between childhood obesity and the head of household socio-economic factors such as employment. The disparity could be attributed to the study’s location as well as the small sample size. In addition, the educational level of the head of the household was found not to be statistically associated with childhood overweight. This contradicted previous findings, which stated that parents’ educational level might contribute to their children’s nutritional knowledge, which may lead to the development of obesity in their children [28,33,34]. This might be because the current study had a majority (88%) of the heads of the household who completed less than high school, which could have led to a discrepancy in the study findings.

### Study Limitations

The study included the following possible limitations: First, the study was a cross-sectional design, preventing the assertion of a causal association between anthropometric indices of adiposity and blood pressure. The second factor is the small patient population.

## 5. Conclusions

Obesity in children has been attracting increasing attention in rural and urban areas because of concerns about its long-term effects on adulthood. The present study presents a rare insight into rural-dwelling black South African children and adolescents who are overweight and the influence of household head socioeconomic status on their overweight. The results of the present study suggest that variables such as household head gender, education level, employment status, religious faith, and relationship with the child are not associated with child overweight in rural areas. Children may inherit genetic predispositions from their parents. In an area dominated by cardiovascular risk factors (i.e., overweight or obesity and hypertension), screening and monitoring children’s anthropometric indices and blood pressure regularly is needed for the effective prevention and management of chronic diseases. We further propose that public health interventionists be targeted both at heads of household and children in hopes of reducing the prevalence of obesity in South Africa’s children.

## Figures and Tables

**Figure 1 children-09-01728-f001:**
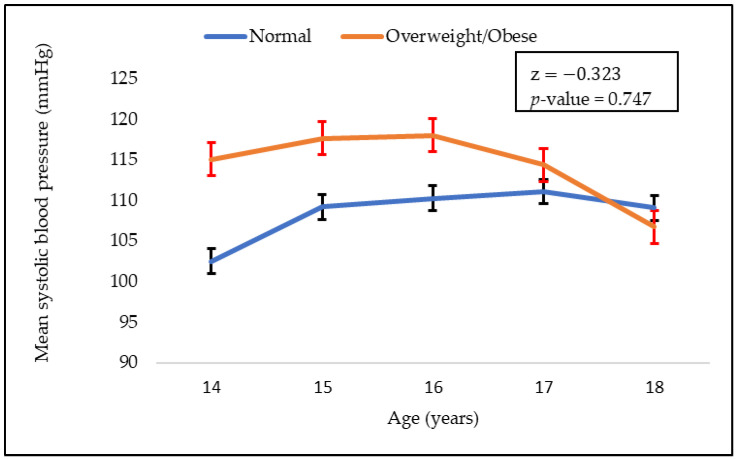
Adolescent’s mean systolic blood pressure values according to their BMI group (normal and overweight).

**Figure 2 children-09-01728-f002:**
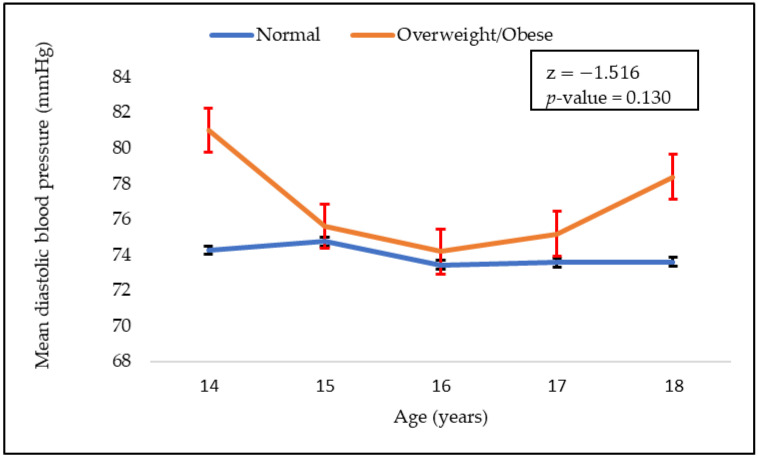
Adolescent’s mean diastolic blood pressure values according to their BMI group (normal and overweight).

**Table 1 children-09-01728-t001:** Characteristics of the participants by BMI group (normal and overweight).

	Total (%)	Normal Weight (%) Adolescents 127 (43)	Overweight (%) Adolescents 167 (57)	Chi-Square *p*-Value
Child’s (gender)				<0.001 *
Girl	154 (52)	106 (69)	48 (31)	
Boy	140 (48)	131 (94)	9 (6)	
Social grant				0.001 *
No	48 (16)	30 (62)	18 (38)	
Yes	246 (84)	207 (84)	39 (16)	
Head of household (gender)				0.943
Female	200 (68)	161 (81)	39 (19)	
Male	94 (32)	76 (81)	18 (19)	
Age group				0.016 *
19–34	103 (35)	81 (79)	22 (21)	
35–44	59 (20)	53 (90)	6 (10)	
45–54	52 (18)	35 (67)	17 (33)	
55+	80 (27)	68 (85)	12 (15)	
Financial status				0.005 *
Comfortable	29 (10)	22 (76)	7 (24)	
Extremely poor	12 (4)	12 (100)	0 (0)	
Just getting by	183 (65)	156 (85)	27 (15)	
Poor	56 (20)	30 (64)	26 (36)	
Education level				0.485
Above high school	29 (10)	24 (83)	5 (17)	
High school	5 (2)	3 (60)	2 (40)	
Less than high school	260 (88)	210 (81)	50 (19)	
Employment status				0.802
No	239 (81)	192 (80)	47 (20)	
Yes	55 (19)	45 (82)	10 (18)	
Religious faith				0.716
No	53 (24)	46 (87)	7 (13)	
Yes	164 (76)	139 (85)	25 (15)	
Head of household overweight				<0.001 *
No	127 (43)	117 (92)	10 (8)	
Yes	167 (57)	120 (72)	47 (28%)	
Relationship				0.601
Parents	78 (27)	65 (83)	13 (17)	
Relatives	207 (70)	164 (79)	43 (21)	
Caretaker	9 (3)	8 (89)	1 (11)	

* 0.05 significant level.

**Table 2 children-09-01728-t002:** Logistic regression model for the overweight adolescents with selected socioeconomic covariates among the heads of households.

	OR	95%CI	*p*-Value
Head of household: Male	1.411	(0.562; 3.539)	0.463
Child: Boy	0.163	(0.067; 0.396)	<0.001 *
Age group			
19–34	2.364	(0.885; 6.313)	0.086 **
35–44	0.897	(0.243; 3.307)	0.871
45–54	3.720	(1.203; 11.501)	0.023 *
55+	Reference		
Financial status			
Just getting by	0.333	(0.098; 1.127)	0.077 **
Poor	0.830	(0.218; 3.164)	0.785
Comfortable	Reference		
Education status			
High school and above	1.068	(0.353; 3.226)	0.908
Less than high school	Reference		
Unemployed	1.473	(0.490; 4.424)	0.491
Receiving child grants	0.281	(0.112; 0.703)	0.007 *
Head of household overweight	5.757	(2.316; 14.312)	<0.001 *

* Significant at 5% level; ** Significant at 10% level.

## Data Availability

The data presented in this study are available on request from the corresponding author. The data are not publicly available due to privacy or ethical restrictions.

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
