# Peer review of "Overweight Prevalence among Rural Adolescents by Household Head Obesity and Socio-Economic Status in Limpopo, South Africa"

_children, 2022, doi:10.3390/children9111728_

Round 1
Reviewer 1 Report
This study could not confirm the differentiation from previous studies on the prevalence of overweight and obesity among rural adolescents. It is believed that the originality of this study can be recognized only when the relationship between the characteristics of South Africa's Limpopo region and the prevalence of overweight and obesity among adolescents must be added to the theoretical background.
Additionally,
line 52 : Reference number?
line 55 : . Such
line 64 : Need to delete one extra space
line 89 : No comma needed here
line 96 : Reference number?
line 121 : ;
line 122 : Therefore,
line 124 : overweight
line 133 : Need to delete one extra space.
line 139 : Need to delete one extra space.
line 171 : overweight
line 175 : overweight
Table 1. Social Grand -> Social Grant
line 206 : No comma needed.
line 222 : ,
line 224 : and indicated that boys have~
line 224 : situation
line 225 : adults
line 225 : which women have
line 226 : the study -> (deleted)
line 246 : overweight
line 249 : overweight
line 250 : In an area dominated by cardiovascular risk factors(i.e., overweight or obesity, and hypertension), screening and monitoring children’s anthropometric indices and blood pressure regularly is needed for effective prevention and management of chronic diseases.

Author Response
Thank you very much for the support and the comments on our manuscript, which do help us significantly improve the quality of the current article.
Thank you for your note. The authors tried to address all the comments and again the article was taken to the language editor for grammar and spell checks.
Reviewer 2 Report
Congratulations on the study. I have suggestions that aim only at the greater appreciation of the results presented and a more consolidated discussion.
My first suggestion is that the summary has a more assertive conclusion. The summary ends very vaguely.
Forgive me if I missed my perception, but I didn't understand if there were 294 or 295 participants.
In my opinion, the discussion of the article has to be improved. The introduction, which was very well written, justified the topic's relevance and explained what the development of the study adds to knowledge.
In the discussion, there should be further exploration and interpretation of the results. I think there should be more arguments about: the cardiovascular parameters measured, the possible causes for the differences in weight between girls and boys, the influence of the child grant, and the differences between the rural population studied, and the urban population evaluated in previous studies. The results have potential and may raise important questions for future studies.
Author Response
Thank you very much for the support and the comments on our manuscript, which do help us significantly improve the quality of the current article.
Forgive me if I missed my perception, but I didn't understand if there were 294 or 295 participants.
Thank you for your note. The authors corrected that in the document.
In my opinion, the discussion of the article has to be improved. The introduction, which was very well written, justified the topic's relevance and explained what the development of the study adds to knowledge.
In the discussion, there should be further exploration and interpretation of the results. I think there should be more arguments about: the cardiovascular parameters measured, the possible causes for the differences in weight between girls and boys, the influence of the child grant, and the differences between the rural population studied, and the urban population evaluated in previous studies. The results have potential and may raise important questions for future studies.
Thank you for your note. The authors tried to improve the discussion section.
Round 2
Reviewer 1 Report
You suffered.